# Hallmarks of Bacterial Vaginosis

**DOI:** 10.3390/diagnostics15091090

**Published:** 2025-04-25

**Authors:** Diana Cristina Pérez-Ibave, Carlos Horacio Burciaga-Flores, Ximena García-Mejía, Fernando Alcorta-Nuñez, Orlando Solis-Coronado, Moisés González Escamilla, Oscar Vidal-Gutiérrez, María Lourdes Garza-Rodríguez

**Affiliations:** 1Servicio de Oncología, Centro Universitario Contra el Cáncer (CUCC), Hospital Universitario “Dr. José Eleuterio González”, Universidad Autónoma de Nuevo León, Monterrey 66451, Mexico; dperezi@uanl.edu.mx (D.C.P.-I.); bufc1655389@uanl.edu.mx (C.H.B.-F.); fernando.alcortannz@uanl.edu.mx (F.A.-N.); orlando.solisc@gmail.com (O.S.-C.); moises.gonzalezes@uanl.edu.mx (M.G.E.); 2Facultad de Medicina, Universidad Autónoma de Nuevo León, Monterrey 66451, Mexico; ximena.garciamj@uanl.edu.mx

**Keywords:** bacterial vaginosis, dysbiosis, hallmarks, vaginal microbiota, Lactobacillus, *Gardnerella vaginalis*, vaginal epithelial damage

## Abstract

**Background:** Bacterial vaginosis (BV) is considered the most common cause of vaginal discharge, which is related to several public health issues, such as an increased risk for sexually transmitted infections, pelvic inflammatory disease, pregnancy-related problems such as abortion, stillbirth or premature birth, and tubal factor infertility. BV is not considered an infection but an imbalance in the vaginal microbiota, characterized by a substitution of the normal Lactobacilli flora by anaerobe. Reducing resistance against infections by several mechanisms, including bacterial homeostasis, stabilization of acid pH, inhibition of pathogens adhesion by polyamine degradation, production of anti-inflammatory molecules, surfactants, and antimicrobial substances like hydrogen peroxide, acids, and bacteriocins. Approximately half of women with BV can experience symptoms, which mainly include vaginal malodor, fishy discharge, stinging sensation, and increased vaginal pH. The treatment of BV is based primarily on promoting Lactobacilli restoration and eliminating dangerous microbiota with antibiotic therapy. However, there is a high rate of recurrence and relapse. **Objective:** Based on the current literature, this review aims to propose a list of ten BV hallmarks: dysbiosis, inflammation, apoptosis, pH basification, mucosal barrier integrity, pathway activation, epithelial damage, genomic instability, oxidative stress (OS), and metabolic reconfiguration. **Conclusions:** Understanding the causes of BV and the pathogenicity mechanisms is critical for preventing and improving the current therapeutic management of patients.

## 1. Introduction

The cervicovaginal epithelium is a transition epithelium with a wide range of defenses for infections, including (a) mucus production as a physical barrier, (b) maintenance of microenvironment, (c) control of cell proliferation, (d) immunological response, (e) cytokine expression, and (f) proinflammatory signaling [1].

Normal vaginal microbiota is composed mainly of Lactobacillus species (*L. crispatus*, *L. gasseri*, *L. iners*, and *L. jensenii*). These bacteria protect the epithelium by creating a healthy acidic microenvironment with antimicrobial compounds, a competitive exclusion from foreign pathogens, and other mechanisms [2]. Risk factors for BV include vaginal douching, starting sexual life at a young age, multiple sexual partners, the practice of sex work, antibiotics, smoking, high-stress levels, and the use of intrauterine devices [3].

Abnormal conditions in the vaginal microbiota lead to Bacterial Vaginosis (BV). BV is an alteration of the normal vaginal flora that involves a decrease in protective Lactobacilli and an overgrowth of polymicrobial anaerobic bacteria [4,5,6,7,8]. The factors that trigger anaerobic bacteria overgrowth are not well established, but it is known that vaginal microenvironment alkalization through pH basification and Lactobacilli substitution reduces the innate protective effects [9]. Nowadays, BV is considered a condition rather than a sexually transmitted infection (STI), but it is an important risk factor for acquiring one [5,10,11,12].

BV worldwide prevalence is around 20% to 60% of women. African regions have the highest prevalence, while Asia and Europe have the lowest [11]. In the United States, the mean rate of BV is 30%, with a differential in non-white women: African American women have a prevalence of 51%, and Mexican American women have 32% [13,14]. In Mexico, the prevalence is 32% among the general population, but in sex workers, this can be as high as 57% [13,15].

BV etiology includes an increase of foreign pathogens, with *Gardnerella vaginalis* as the most frequently associated pathogen, followed by *Atopobium vaginae*, *Megasphaera types*, *Leptotrichia amnionii*, *Sneathia sanguinegens*, *Porphyromonas asaccharolytica*, a bacterium related to *Eggerthella hongkongensis*, *Prevotella* spp., *Peptostreptococcus* spp., *Aerococcus*, *Anaerococcus*, *Gemella*, and *Veillonella* spp. [8]. There is also a decrease of typical lactobacillus species like *L. fermentum*, *L. gasseri*, *L. iners*, *L. jensenii*, *L. mucosae*, *L. ruminis*, *L. salivarius*, and *L. coleohominis* from 35% to 6% [16].

Clinically, BV is characterized by a pale, greyish fishy-smelling vaginal discharge as the main symptom and can be related to dysuria, dyspareunia, vaginal pruritus, perianal irritation, edema, and the absence of leukocytic exudate. Still, almost half of the patients do not present any symptoms [17,18].

Amsel’s criteria use four characteristics to diagnose BV, including characteristic pale discharge, vaginal pH higher than 4.5, presence of clue cells by microscopy, and foul odor. BV diagnosis is made when patients have at least three criteria. Another method to diagnose BV is to use the Nugent score, a differential between normal Lactobacillus flora and Gram-variable microorganisms. More recently, the FDA-approved molecular diagnosis tests for BV, including qPCR and sequencing for *Gardnerella* spp., *F. vaginae*, and *Lactobacillus* spp., and FISH for detection of 16S rRNA of the bacterial cells [19,20,21,22]. *G. vaginalis* is the most sensitive indicator of BV, as it has been identified in almost 100% of cases [23].

BV is a risk factor for STIs caused by various pathogens, including viruses, bacteria, and protozoan parasites [2]. It has been demonstrated that patients with BV have a 1.53-fold higher risk of developing pelvic inflammatory disease (PID) and face an increased risk of pregnancy-related complications, including a 6.32-fold higher risk of miscarriage and a 2.16-fold higher risk of preterm birth. Additionally, there is a 3.32-fold higher risk of infertility [22]. Normal vaginal microbiota help prevent HPV infection and accelerate its clearance [4,24,25]. There are some clues that the pro-carcinogenic cervicovaginal microenvironment is caused by dysbiosis, but it has not been completely elucidated [26]. It is known that there is a combination of alterations in several hallmarks of cancer, including barrier disruption by epithelial damage via proteolytic enzymes, predisposing to HPV infection, and abnormal cellular proliferation, like in *Fusobacterium* spp. This process activates the WNT signaling pathway by Fad A virulence factor commonly overexpressed in cervical cancer. There is also genomic instability due to ROS (Reactive Oxygen Species) production, which leads to double-strand breaks in the HPV episome and host genome, which increases the risk for HPV integration and tumor promotion. Angiogenesis and chronic inflammation may increase the carcinogenic potential of HPV and inhibit apoptosis via E6 and E7 viral proteins. Finally, there is a metabolic dysregulation with the production of organic acids (3-hydroxybutyrate, eicosenoate, and oleate) found in patients with cervical neoplasia; also, changes in metabolites from lactobacilli, including an increase in biogenic amines and glycogen-related, products, and a decrease in glutathione, glycogen, and phospholipids [27].

Treatment of BV aims to eliminate dysbiosis and restore the vaginal microenvironment homeostasis [28]. Initial treatment has a cure rate of 80% to 90% within the first month, and one-third of patients resolve without treatment [29].

Clindamycin or metronidazole is the standard of care for symptomatic patients because of its activity against Gram-variable bacteria and protozoa [11,30]. However, over half of the treated patients experience a recurrence in less than 12 months [28]. BV recurrence can be due to multiple causes, including re-exposure to BV-associated pathogens, ineffective recolonization by Lactobacilli, antimicrobial resistance due to the expression of antimicrobial resistance genes, and biofilm formation [31,32]. Biofilms are composed of an assemblage of diverse microorganisms adhered to a surface. Therefore, biofilms constitute a mechanism of antimicrobial resistance, as they reduce the penetration rate of antibiotics into the infection site, leading to a decrease in therapeutic efficacy. These structures represent an initial event for BV’s development and progression [32,33]. The biofilm function as a mechanism of resistance vs. the normal protective vaginal microbiota (hydrogen peroxide and lactic acid), which are generated by Lactobacilli [33].

Modification of treatment, such as an extension of antibiotics, a combination of oral and vaginal antibiotic therapy, and adjuvant probiotic treatment, have no conclusive evidence of long-term treatment success [34]. This is why newer therapeutic options such as antiseptics, DNases, synthetic antimicrobial peptides (retrocyclines), and plant-derived compounds are being developed for BV treatment. Recently, a cell wall-degrading enzyme (endolysin PM-477) showed a specific effect for *Gardnerella* spp. in vaginal discharge samples of BV patients [35]. Also, treatment of the partner is crucial because having an untreated partner increases 2–3 times the risk of recurrence, and the incidence of BV increases with the change of sexual partner [29].

BV is a significant health issue, with annual spending of USD 4.8 billion in the USA only for symptomatic cases; this could be doubled if asymptomatic patients were included [36]. The presence of asymptomatic BV patients still troubles physicians, as those women that will eventually become symptomatic and can develop essential sequelae, including malignant neoplasia [23].

This review aims to summarize the features of BV, propose a list of ten hallmarks of this condition (dysbiosis, inflammation, apoptosis, pH basification, mucosal barrier integrity, pathway activation, epithelial damage, genomic instability, OS, and metabolic reconfiguration), and raise awareness for prevention, treatment, and related complications (Figure 1).

## 2. Dysbiosis and pH Basification

Normal vaginal eubiosis is characterized by specific bacterial flora, regulated acid pH levels, and other factors that contribute to a healthy microenvironment. Dysbiosis is defined as an imbalance of bacterial communities [8]. In BV, the most common anaerobes that generate dysbiosis include *G. vaginalis*, *Mobiluncus* spp., *Prevotella* spp., *Fannyhessea vaginae*, and others [37].

The acidic pH of the vaginal microenvironment is part of the homeostasis mechanisms, maintained by lactic and acetic acid production, repressing pathogen growth. Dysbiosis produced by microorganisms like *G. vaginalis* and *Candida albicans* elevates vaginal pH making a less acidic (pH > 4.5) microenvironment by proteolytic enzymes and decarboxylases that degrade proteins and amino acids; this alkalinization leads to a decrease of Lactobacilli by competitive exclusion and contributes to yeast colonization and tissue invasion accessing epithelial cell adherence sites [7,23].

Vaginal dysbiosis increases susceptibility to viral infections, genital inflammation, and cervical neoplasia. Watts and colleagues found an association between abnormal vaginal flora and HPV prevalence but no association for HPV infection or increased risk for squamous intraepithelial lesions transition [38]. However, other studies found dysbiosis by Fusobacterial (*F. nucleatum*, *Fusobacterium necrophorum*, and *Sneathia* spp.) alters homeostasis and has been found in cervical cancer. Some studies also found these pathogens in endometrial cancer samples [24,25,26] (Figure 2).

## 3. Inflammation

The female reproductive tract is regulated by sexual hormones and microbiota linked to immunity (innate and adaptive) [39]. The vaginal epithelium contains various immune elements, including antibodies, dendritic cells, macrophages, mucosal immunoglobulin A (IgA), T cells, B cells, and epithelial surveillance nucleotide-binding oligomerization domain and receptor-like dectin-1, as well as pattern recognition receptors (PRR) (such as Toll-like receptors) [40,41]. Under normal conditions, the expression of chemokine C–C motif ligand 5 (CCL-5), interleukins (ILs), macrophage inflammatory protein (MIP-1β, MIP-3α), RANTES, secretory leukocyte protease inhibitors (SLPIs), TNF-α, and FMS-like tyrosine kinase-3 ligands is low [42]. Inflammatory changes in BV increase susceptibility to STIs, including a wide range of pathogens [41]. Chronic mucosal inflammation is characteristic of BV due to microbiome dysregulation and depletion of anti-inflammatory molecules present in the vaginal microenvironment [43]. L-lactic acid isomer produced by Lactobacillus also has a role in the immune response by T-lymphocyte pathway stimulation and activation. Vaginal epithelial cells generate proinflammatory cytokines in the presence of viral RNA, which leads to suppression of bacterial growth [44].

Vaginal fluid samples of patients with BV have an increase of PRR that initiates cytokine/chemokine signaling cascades, with IL-8 as a main chemotactic and activating factor for neutrophils. However, IL1α, β, and Ɣ have also been found with other factors like interferon-c-induced protein IP-10, TNF-α, and granulocyte-macrophage colony-stimulating factor (GM-CSF) [27,40,41,45,46,47]. IL-1β and IL-10 are responsible for impairing cytotoxic T-cell response related to HPV persistence and cervical neoplasia [48]. Also, there are other factors triggered by the inflammatory response, including Mannose Binding Lectins, defensins, immunoglobulins, and antimicrobial peptides, all of which maintain vaginal eubiosis [39]. *Atopobium vaginae* activates the nuclear factor-kB (NF-kB), IL-6, IL-8, MIP-3α, and TNF-α [27].

Some specific pathogens can generate differential inflammatory responses, including *Chlamydia trachomatis*, which causes cytokine-mediated inflammatory responses, generating epithelial changes and promoting HPV oncoprotein production [48]. *G. vaginalis* increases IL-8 and IL-1β [41,49,50,51]. *Megasphaera elsdenii* and *Prevotella timonensis* activate dendritic cells and stimulate cytokine production (IL-1β, IL-6, IL-8, IL-12A + IL-12B, and TNFα) [48]. In the colonic mucosa, levels of *Fusobacterium nucleatum* are inversely proportional to CD3+-T cells. This has also been observed in cervical cancers and relapsing disease [52]. These specific pathogen cases reflect the heterogenicity of BV and, in some cases, the lack of symptoms in patients, so detection of these proinflammatory biomarkers (IL- 1a, IL-1b, and IP-10) can be a cheap and easy way to identify women at risk of developing STI, to provide an appropriate treatment and help in cervical cancer prevention [42].

The NOD-like receptor family pyrin domain-containing 3 (NLRP3) is activated by *G. vaginalis*. The NLRP3 inflammasome (NLRP3, apoptosis-associated speck-like protein [ASC], and procaspase-1) drives the production of proinflammatory cytokines (TNF-α, IL-1β, and IL-18), ROS, and initiates cell death by pyroptosis, a mechanism contributing to BV pathogenesis [41,53,54].

Innate immunity could explain differences in response to the infection and progression to sequelae in BV patients. Further genetic analysis could enlighten physiopathology, provide personalized medicine, and develop newer treatments and vaccines for the benefit of patients, evolving gynecological practice to newer standards [23].

## 4. Oxidative Stress and Apoptosis

OS is an imbalance between ROS production and antioxidant defenses [9,55]. ROS (O_2_-, O_3_-, OH.) and reactive nitrogen species (NOS) (NO, O_2_, ONOO-, O_2_-) are highly unstable molecules known as free radicals [56]. ROS and NOS production are natural by-products of regular cell activity and participate in cellular signaling. ROS are generated by the mitochondrial electron transport chain, cytochrome P450 enzymes, hemeproteins, inducible nitric oxide synthesis (iNOS), nicotinamide adenine dinucleotide phosphate (NADPH), oxidases, peroxisomes, and phagocyte cells [57]. Catalase (CAT) and superoxide dismutase (SOD) convert O_2_- into H_2_O_2_ and H_2_O_2_ into H_2_O and O, respectively [58,59]. OS can be measured by SOD and CAT activity and by levels of ROS and malondialdehyde (MDA), a product of unsaturated fatty acid oxidation [58]. Other key metabolites related to OS include 2-hydroxyglutarate and 2-hydroxybutyrate [2].

In vaginal eubiosis, lactobacilli produce H_2_O_2_ as a mechanism of protection against many microorganisms. In BV, the anaerobic and aerobic flora overgrowth is due to the reduced number of lactobacilli, which dysregulates ROS production and, therefore, standard protection [60]. ROS are related to essential cell functions such as proliferation, hypoxia adaptation, and programmed cell death. Overproduction of these metabolic subproducts leads to cell damage, genome instability, apoptosis, protein, and lipid oxidation that impact cell membrane fluidity [9,60,61] (Figure 3).

Mojgan T. et al. addressed the relationship between OS and an increased risk of developing BV by depleting antioxidant agents that lead to ROS overproduction and inflammation [9]. Analysis of BV vaginal discharge samples has found increased levels of H_2_O_2_ and MDA and depletion of CAT enzyme activity and vitamin C compared to healthy individuals [58,60]. Overgrowth of pathogens that substitute *Lactobacillus* spp. with anaerobic organisms such as *Bacteroides* spp., *Fusobacterium* spp., *Mobiluncus* spp., *Mycoplasma hominis*, *Peptostreptococcus* spp., *Prevotella* spp., *Prophyromanas* spp., and Gram-variable microorganisms, such as coccobacilli and *G. vaginalis*, in the vaginal microenvironment promote oxidative damage by H_2_O_2_ and MDA overproduction [60,62].

OS-damaging properties increase apoptotic signals, including the release of cytochrome C into the cytoplasm by changes in mitochondrial membrane permeability, leading to caspase cascade activation. BCL-2 proteins control mitochondrial membrane permeability and act as an agonist for apoptosis; meanwhile, Bax acts as an antagonist to cytochrome C release. Zhen Zhang et al. found a significant decrease in BCL-2 BV epithelial cell samples and an increase in BAX [60]. Other studies have shown increased cytochrome C and caspase-3 levels in BV samples. In particular, *G. vaginalis* can induce its activation and is thought to be related to an increase in apoptotic response in BV [63,64]. Experimental treatments have included Vitamin C in the standard metronidazole treatment to protect the epithelium from OS [60]. Vitamin C administration seems to improve BV symptoms and restoration of normal vaginal microenvironment (bacterial and lactobacilli equilibrium and pH acidification) [55,65].

OS plays an essential role in HPV infection and cancer development by creating DNA breaks in the host genome and changes in the episomal status of HPV. This genomic change facilitates viral integration and cell transformation, increasing the risk of neoplasia [52].

## 5. Genomic Instability and Pathway Activation

Among the multiple features present in BV, dysbiosis can promote chronic bacterial infections. Some pathogens’ subproducts include cytolethal-distending toxin (CDT) that arrests the cell cycle in G1 or G2 and colibactin, a cyclomodulin toxin that induces chromosome instability and DNA damage. In addition to these products, the bacterial low-fidelity DNA repair mechanisms (Non-Homologous End-Joint Repair and translesion repair) generate a mutagenic effect that can influence the adaptation of the epithelium but also pose a potential risk for genetic instability [66,67].

BV disruption promotes HPV oncogenesis. Reports suggest that lipopolysaccharides produced by anaerobic bacteria dysregulate oncogenic drivers like p53, pRB, survivin, hTERT, and E-cadherin, which are also disrupted by HPV oncoproteins. There is evidence that some bacteria in the cervical epithelium, like *Escherichia coli*, produce colibactin and, in conjunction with HPV-16, can promote CIN [68]. Despite this, there is no clear evidence of a synergy between the bacteria and HPV or if the bacteria act as an independent factor for neoplasia; however, there is evidence that some specific bacteria have a role in the acquisition and persistence of HPV infections, but there is no proof that dysbiosis can increase these risks by itself [52].

Another feature of BV that can promote a neoplastic transformation in the host is the activation of growth pathways. It is known that HPV by itself can be carcinogenic, but some bacterial co-infections contribute to this process via the secretion of activating products. *Fusobacterium* spp. produces FadA, which can activate the WNT pathway, which has been found to be dysregulated in cervical cancer [52]. Mycoplasmas and Ureaplasmas secrete DnaK, a protein that reduces PARP1 and p53 activity, thus contributing to cellular transformation by hampering p53 activity and promoting chromosomal instability [69].

## 6. Metabolic Reconfiguration

Every tissue has a specific metabolism related to its activity. In the human vagina, these metabolites are a result of the interconnection between the bacterial metabolisms and the host nutrients. One of the main nutrients precursors in the vaginal environment is the glycogen used by *Lactobacillus* spp. as maltose. BV decreases glucose and glycolysis and also dysregulates lipids and amino acid metabolism, which, in turn, can impact the energy metabolism of epithelial cells. There is also a reduction in carbohydrate metabolism, which is reflected in amino acid depletion and energy metabolites (NAD+). The bacterial flora impacts homeostasis, leading to abnormal cell function, inflammation, and disease susceptibility [70,71,72].

BV has at least 176 different dysregulated metabolites (subproducts of glucose metabolism, ammonia, among others) useful as biomarkers [2,44,70,71,73,74]. Metabolic changes in BV include disruption of multiple cellular metabolites, starting from proteins with increased amines (cadaverine, trimethylamine, and tyramine), giving the characteristic fishy odor. Also, there is a decline in all amino acid levels except for proline [71,72]. Short-chain fatty acids tend to increase in BV, possibly affecting the activation of innate immune cells [75]. Finally, there is a dysregulation in organic acid production with increased propionate and acetate, carboxylic acids such as acetate, malonate, and niacin, also a decrease in the normal lactic acid production by Lactobacillus [71,72]. Organic acids and polyamines in BV cause exfoliation of vaginal epithelial cells, which leads to abnormal vaginal discharge [30].

As said before, persistent HPV infection can be promoted by BV. There is evidence of metabolic changes that can occur in BV but also in patients with HPV-related dysplasia and cervical cancer. In HPV-related low and high-grade dysplasia and cervical cancer, there is a depletion of amino acids (taurine, glutamine, and lysine), dipeptides, and an increase of lipids (3-hydroxybutyrate, long-chain polyunsaturated fatty acids, phosphatidylcholines, plasmalogens, and sphingomyelins) and xenobiotics. It has been reported that taurine and glutamine are related to cell growth, and their disruption can lead to tumorigenesis [27,76]. In cervical cancer samples, there is an increase in biogenic amines and glycogen metabolism and a decrease in phospholipid metabolites [77].

## 7. Mucus Disruption

Mucus covers and protects the epithelial surfaces of the female genital tract, which is produced by the endocervical canal. It is comprised mainly of mucins and water, also including other metabolites (carbohydrates, immunoglobulins, inorganic ions, lipids, plasma proteins, and sterols), which are part of the physical barrier of the epithelium, trapping foreign pathogens (bacteria, viruses, and fungi) [78,79]. Among all the other elements, cervical mucus differs from other mucosal epithelium by the influence of sex hormones that regulate the quantity and qualities of the mucus, and this has been demonstrated in hormonal transition periods such as menarche, menses, pregnancy, and menopause [80,81]. There are 20 mucins (gels, transmembrane proteins, and small soluble molecules). The main cervical mucins are MUC5AC and MUC5B, serving as the barrier against STIs. In addition to mucins, neutrophils play a critical role in mucosal homeostasis as they protect from infections. They can also lead to inflammation and potential tissue damage by increased proteases with inflammatory cytokines [78]. Also, in normal vaginal mucus, sialic acid contributes to this physical protection, which is depleted in BV patients [82].

Mucus degradation plays an essential role in BV and its complications. This event has been observed in BV by the presence of glycosidases in vaginal fluid [31,32,33]. Even though numerous bacterial species have demonstrated hydrolysis and depletion of mucosal sialic acids, *G. vaginalis* is sufficient to cause mucosal sialic acid depletion [82].

## 8. Epithelial Damage

The normal vaginal epithelium is the first barrier against STI with various cell protective mechanisms like mucus-secreting cells, suppression of de-keratinization, immune cell stimulation, control of cell proliferation and shedding [1,83,84]. *Lactobacillus* spp. contributes to standard vaginal protection by multiple mechanisms, with physical and immunological defenses as some of the most important. BV is a known factor of epithelial disruption, affecting cell polarity, diffusion control, and cell junctions. These changes increase permeability, facilitating the entry of pathogens and toxins [84] (Figure 2).

Cervicovaginal microbiota in BV patients shed twice as much as those found in healthy women. The accelerated shedding of cells from vaginal epithelium increase susceptibility to STIs and cervical dysplasia [1,85]. Also, cell maturation and proliferation are decreased in BV patients due to the overexpression of miR-193b by abnormal vaginal flora [1].

In BV, there is a modification in the glycan; the main example is the production of glycosidic enzymes by *G. vaginalis*, altering epithelial gene expression, accelerating cell turnover, microbiome response, and cytoskeletal reconfiguration [83,86]. Other external non-bacterial products associated with epithelial disruption are topical products and physical disruption by trauma [87].

## 9. Conclusions

BV is a common genital condition that affects the vaginal microbiome homeostasis. Internal and external factors contribute to disrupting the balance in the vaginal ecosystem (dysbiosis). The vaginal microbiome has a vital role in homeostasis.

We still need to understand and clarify how the interaction between the BV hallmarks and the host impact vaginal health. Most of the women with BV require treatment, and there is a high relapse rate. BV recurrence has been linked to a detrimental stage in social, sexual, and emotional health; it diminishes the quality of life of affected women.

There is also a need for more evidence to elucidate whether dysbiosis is correlated with HPV persistence and cervical cancer.

In this review, we propose 10 hallmarks that define BV: dysbiosis, inflammation, apoptosis, pH basification, mucosal barrier integrity, pathway activation, epithelial damage, genomic instability, oxidative stress (OS), and metabolic reconfiguration. These hallmarks lead to a better understanding of BV and could improve diagnosis, prevention, and treatment strategies to avoid possible complications.

## Figures and Tables

**Figure 1 diagnostics-15-01090-f001:**
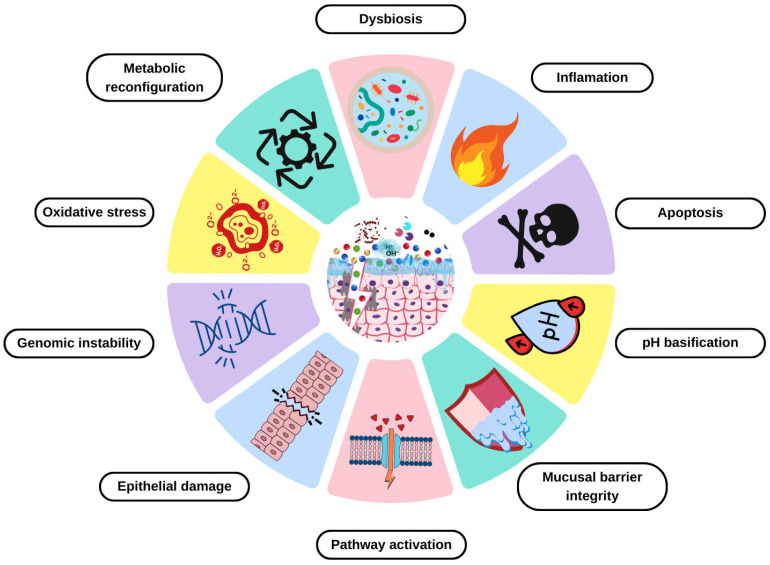
Hallmarks of BV. This illustration encompasses the ten proposed BV hallmarks (Canva V 1.102.0).

**Figure 2 diagnostics-15-01090-f002:**
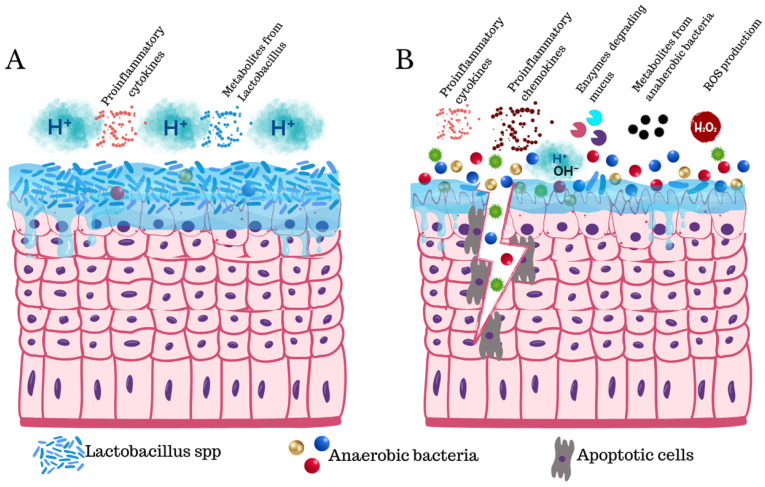
Normal vaginal epithelium and BV microenvironment changes. (**A**) Vaginal eubiosis characterized by healthy vaginal flora, normal mucus production, and a stable acidic pH level. (**B**) Microenvironmental changes in the BV epithelium, including epithelial damage, mucosal barrier degradation, an increase in proinflammatory signaling, pH increase, ROS, and proapoptotic factors production (Canva V 1.102.0).

**Figure 3 diagnostics-15-01090-f003:**
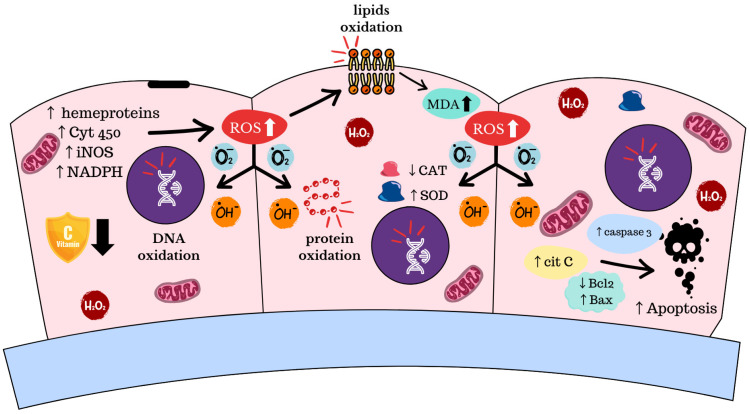
OS homeostasis disruption induced by ROS overproduction in cervicovaginal epithelium in BV conditions (Canva V 1.102.0).

## Data Availability

Not applicable.

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
