# Peer review of "Hallmarks of Bacterial Vaginosis"

_diagnostics, 2025, doi:10.3390/diagnostics15091090_

Round 1
Reviewer 1 Report
Comments and Suggestions for Authors
Manuscript reference number: diagnostics-3556422
Journal: diagnostics
Title: Hallmarks of Bacterial Vaginosis
Authors: Pérez-Ibave et al.
Reviewer’s Comments/recommendations:
This review highlights Bacterial Vaginosis (BV), an imbalance of vaginal microbiota characterized by grayish-white discharge, high vaginal pH (greater than 4.5), presence of clue cells by microscopy (Wet mount), and fishy odor (positive Whiff test). This is an important vaginal condition affecting more than a third of women.
The authors propose a list of ten hallmarks of BV: dysbiosis, inflammation, apoptosis, pH basification, mucosal barrier integrity, pathway activation, epithelial damage, genomic instability, oxidative stress (OS), and metabolic reconfiguration. It is important to understand the causes of BV and the pathogenicity mechanisms that is critical for preventing it and improving the current therapeutic management of patients.
This review is generally well-structured and discusses important and more comprehensive factors that contribute to better understanding of BV condition.
Here is a list of text corrections for authors to do:
Line 14: … is considered is as the most common ….
Line 18: replace misbalance with imbalance.
Line 70: add “,” after clinically, …
Line 75 replace the wrong phrase “low pH (below 4.5)” with the correct one “vaginal pH higher than 4.5” (they have it correct on line 147).
Line 104: add “vaginal” before microenvironment
Line 122: change antimicrobian .. to antimicrobial ..
Line 124: change “an specific” to “a specific”
Line 134: change “infection” to “condition”
Line 191: … observed in cervical cancers … (add “in”)
Line 349: Correct sentence to read: BV is a common genital condition that affects the vaginal microbiome homeostasis.
Line 350: … to disrupting …
Line 361: … hallmarks lead to …
Under Abbreviations: Replace hidrogen with hydrogen. Replace oxigen with oxygen.
I find the article suitable for publication in diagnostics. Thank you.
Overall Recommendation: Accept after minor revisions
Comments on the Quality of English LanguageThe English could be improved. English corrections list is given above under suggestions to authors.
Reviewer 2 Report
Comments and Suggestions for Authors
This review paper entitled Hallmarks of Bacterial Vaginosis, discusses the hallmarks of bacterial vaginosis (BV), a common cause of vaginal discharge linked to public health issues like increased risk of STIs and pregnancy complications. BV is characterized by an imbalance in vaginal microbiota, with a decrease in protective Lactobacilli and an increase in anaerobes.
The paper proposes ten hallmarks of BV: dysbiosis, inflammation, apoptosis, pH basification, mucosal barrier integrity, pathway activation, epithelial damage, genomic instability, oxidative stress, and metabolic reconfiguration. These hallmarks contribute to the pathogenicity of BV, affecting various cellular processes and potentially increasing the risk of HPV infection and cervical neoplasia.
Current treatments for BV involve antibiotics, but recurrence rates are high. The review emphasizes the need for understanding the causes and mechanisms of BV to improve prevention and treatment strategies, and to reduce related complications.
Reviewer 3 Report
Comments and Suggestions for Authors
I consider the article is well-realized and organized. The content is appropriate, and the ideas are correct and presented fluently. I like also the fact that the illustrations are included in order to have a more clearer understanding of scientific content. Congratulations!
I would have some comments:
Delete the 341 line space.
"Some studies address the relationship between OS and an increased risk of devel- 242
oping BV by depleting antioxidant agents that lead to ROS overproduction and inflam- 243
mation [9]." Chang the "some studies" with specific studies name of first author, for example, "The study conducted by reveals".
Try to not let empty space between the paragraphs.
